# Compared Inhibitory Activities of Tamoxifen and Avenanthramide B on Liver Esterase and Correlation Based on the Superimposed Structure Between Porcine and Human Liver Esterase

**DOI:** 10.3390/ijms252413291

**Published:** 2024-12-11

**Authors:** Hakseong Lim, Sungbo Hwang, Seung-Hak Cho, Young-Seok Bak, Woong-Suk Yang, Daeui Park, Cheorl-Ho Kim

**Affiliations:** 1Department of Biological Science, Sungkyunkwan University, Suwon 16419, Republic of Korea; hakseonglim@naver.com; 2Division of Advanced Predictive Research, Center for Biomimetic Research, Korea Institute of Toxicology, Daejeon 34114, Republic of Korea; sbhwang@kribb.re.kr; 3Division of Bacterial Disease Research, Center for Infectious Disease Research, Korea National Institute of Health, Cheongju 28159, Republic of Korea; skcho38@korea.re.kr; 4Department of Emergency Medical Services, Sun Moon University, Asan-si 31460, Republic of Korea; emtsos@hanmail.net; 5National Institute for Nanomaterials Technology (NINT), POSTECH, Pohang 37673, Republic of Korea; yangws91@naver.com

**Keywords:** liver esterase, tamoxifen, avenanthramide B, superimposed structure, molecular docking, simulation, enzyme inhibition

## Abstract

Exposure to tamoxifen can exert effects on the human liver, and esterases process prodrugs such as antibiotics and convert them to less toxic metabolites. In this study, the porcine liver esterase (PLE)-inhibitory activity of tamoxifen has been investigated. PLE showed inhibition of a PLE isoenzyme (PLE5). In addition, avenanthramides, which have a similar structure to that of tamoxifen, have been used to determine the PLE-inhibitory effect. Among the avenanthramide derivatives, avenanthramide B has been shown to inhibit PLE. Avenanthramide B interacts with Lys284 of PLE, whereas avenanthramide A and C counteract with Lys284. Avenanthramide B has shown a similar inhibitory effect to that of tamoxifen. Given that avenanthramide B can modulate the action of PLE, it can be used in pharmaceutical and industrial applications for modulating the effects of PLE. Based on superimposed structures between PLE and human liver esterase, the impact of tamoxifen use in humans is discussed. In addition, this study can serve as a fundamental basis for future investigations regarding the potential risk of tamoxifen and other drugs. Thus, this study presents an insight into the comparison of structurally similar tamoxifen and avenanthramides on liver esterases, which can have implications for the pharmaceutical and agricultural industries.

## 1. Introduction

Carboxylesterases (CES; EC 3.1.1.1) have a history spanning more than 50 years and are categorized as nonspecific esterases that are highly distributed in the liver. CESs, which include acetylcholinesterase, are classified into the serine esterase or serine hydrolase superfamily and are responsible for the metabolism and detoxification of various ester-containing xenobiotics and clinical drugs [1]. Among the five human CES families (CES1–CES5), categorized according to the identity of the nucleotide sequence, the human liver (CES1A1, CE1) and human intestinal forms (CES2, CE2) are functionally important in the detoxication and bioactivation of xenobiotics [2,3,4] due to their ability to hydrolyze esters. In addition to human CESs, various types of porcine CESs are known. Porcine liver esterase (PLE) consists of various CESs that are responsible for the processing of drugs and pesticides, increasing the risk of ecotoxicity [5]. PLE, among the many subsets of esterases, has various applications in both pharmaceutical and industrial aspects [4,5]. Notably, coronavirus relies on esterase activity to detach itself from and exit the host cell [6,7]. PLE can also process prodrugs such as antibiotics; however, information regarding the exact mechanism of its catalytic inhibition is yet to be reported. 

Tamoxifen is a marketed anti-hormonal drug that is widely used to treat breast cancer (BC). The target of tamoxifen is the estrogen receptor (ER), where it acts as an antagonist, thereby inhibiting cancerous BC cells from receiving estrogen. Since its commercialization, tamoxifen has successfully diminished the annual death rate of BC [8]. Tamoxifen belongs to the class of drugs known as selective estrogen receptor modulators (SERMs), which mimic estrogen and bind to estrogen receptors, thereby prohibiting actual estrogens from binding to the receptor [5]. The true blood concentration of tamoxifen can inhibit CESs and cause drug–drug interaction. In addition, there has been no significant correlation between tamoxifen’s inhibition of CESs and hepatotoxicity. Tamoxifen metabolism has been widely reported [9]. Furthermore, tamoxifen metabolism can specifically predict concentrations and outcomes in BC patients [10,11]. Tamoxifen has also been reported to inhibit CES1 with a Ki of 15.2 μM [12,13]. Although tamoxifen has potential beneficial effects on hormonal regulation, it can pose a threat to the liver by inducing hepatotoxicity [14]. Therefore, many animal models including mice, rats, and zebrafish have been utilized to study tamoxifen-induced hepatotoxicity [15]. There are three major SERMs (tamoxifen, toremifene, and raloxifene), and their structures are shown in Figure 1. Tamoxifen and toremifene are similar in that they share a common triphenylethylene structure, whereas raloxifene contains a benzothiophene structure [5]. The tamoxifen-metabolized phenolic compounds such as 4-hydroxytamoxifen show increased human ER-binding activity [16]. Despite its excellent efficacy against ER-positive BC, tamoxifen is known to exhibit various side effects such as hot flashes, menstrual cycle abnormalities, uterine cancer, and hepatotoxicity. Tamoxifen inhibits both human CES1 and rat CES in a partially non-competitive fashion [12]. However, the mechanism behind this molecular inhibition of liver esterases is yet to be elucidated. Currently, known tamoxifen target proteins include human CES1, rat CES, liver fatty acid binding protein, and estrogen [12].

Avenanthramides (Avns) are structurally similar to tamoxifen metabolites and are a major component of oats (*Avena sativa*), which is a cereal crop that is produced and consumed worldwide and has various health benefits. It has recently been identified as a healthy food because of its potentially therapeutic nutritional profile [17]. The phenolic compounds of Avn A, B, and C have various biological roles, including, but not limited to, anti-inflammation, antioxidant, antiatherogenic, and protection against allergens [18,19]. 

In this study, a variety of molecular simulations and physiological assays have been used to determine the effects of tamoxifen and Avns on PLE. A PLE enzyme assay was conducted in accordance with the basic pharmacology and toxicology policy for experimental studies [20]. PLE5 was chosen as a rare form similar to human CES because PLE1 and PLE6 are the most abundant liver CESs in pigs [21]. Tamoxifen, a widely used drug against BC, inhibits the activity of PLE. To explain the molecular interaction between esterase and Avn derivatives, an ab initio quantum mechanical (QM) calculation was performed for the PLE–tamoxifen or PLE–Avn derivatives complex. The QM method obtains information regarding accurate molecular interaction based on wave function from the fragment molecular orbital (FMO) method [22] and pair interaction energy decomposition analysis (PIEDA) [23]. The FMO method is mainly used in biomolecular systems, where the computation cost is very high when using ab initio QM calculations [24,25]. To reduce the computational cost, we introduced the density-functional tight-binding (DFTB) method, which is an efficient semi-empirical QM method and is expected to exhibit reasonable accuracy [26]. The results showed that tamoxifen and Avn B inhibited PLE enzyme activity, known to be similar to human CES.

## 2. Results

### 2.1. Molecular Docking Simulation of Tamoxifen–PLE Interaction

Tamoxifen is a widely used drug that is typically used daily in a hospital environment. Human and pig livers are very compatible with each other at both the structural and cellular levels. In addition, tamoxifen has been reported to potentially induce hepatotoxicity in humans. As shown in Figure 2, tamoxifen binds to the pocket of the active site without Phe286 in HLE. As mentioned in the Introduction, tamoxifen contains a triphenylethylene structure that appears to overlap with Phe286 in the pocket of PLE. The binding site of PLE was determined to be located at the entrance of the active site. The highest docking score for tamoxifen was calculated as −8.6 kcal/mol in HLE and −7.3 kcal/mol in PLE. Although the tamoxifen was weakly bound to PLE, it could be bound to the entrance of the active site in PLE. Therefore, tamoxifen could block the substrate of PLE, thereby acting as a potential PLE inhibitor. Therefore, it can be postulated that tamoxifen also has an inhibitory effect on PLE. An esterase inhibition assay was employed to study the potential inhibitory effect on PLE. As shown in Figure 2, tamoxifen can inhibit the activity of PLE in a dose-dependent manner. Until now, there have been no structural or physiological insights reported on the effect of tamoxifen on PLE. Considering the similarity between the human and pig liver, understanding tamoxifen’s effect on the porcine liver and esterases could be useful in developing a new animal model to study the side effects of tamoxifen.

### 2.2. Molecular Docking Simulation of Avenanthramide B–PLE Interaction

In this study, a molecular docking simulation was performed to determine the structure of the complex of the PLE protein and Avn derivatives. The number of complex structures was predicted to be six poses in Avn A and three poses in the other Avn derivatives by the AutoDock program. The docking scores for the Avn derivatives and root mean square distance (RMSD) from the binding pose are listed in Appendix A. None of the binding poses of the Avn derivatives could be located in the active site of PLE; instead, they were outside of the active site because Phe286 was blocking the entrance to the active site of PLE (Figure 3a). However, the tamoxifen in HLE was bound to the pocket of the active site without Phe286. 

### 2.3. Validation of Docking Poses Between Avenanthramide and B-PLE Using Molecular Dynamics

To validate these binding poses, we performed molecular dynamics simulations using PLE–ligand complexes (Appendix A). If the binding pose is a stable structure, it will exhibit little movement through molecular dynamics. The shortest distances between Phe286 and each ligand were calculated to be below 3.2 Å. This distance could potentially occlude the active site of PLE. In addition, the standard deviations of the RMSD for all the ligands generated by molecular dynamics relative to the initial ligand structure were 0.956 Å (Tamoxifen), 2.357 Å (Avn_A), 2.863 Å (Avn_B), and 1.305 Å (Avn_C). Given that the standard deviations of the RMSDs of all the ligands were calculated to be similar to the 2.4 Å resolution corresponding to the experimentally defined structure of PLE (PDB ID: 5FV4), all the binding poses of the ligands could be regarded as stable poses.

### 2.4. Avenanthramide B–PLE Interaction Analysis Using FMO Calculation

To analyze the interaction pattern between PLE and the Avn derivatives, FMO calculation was performed in this study for all binding poses of the Avn derivatives using the docking simulation results. In a previous study [27], the interaction energy calculated by FMO was highly correlated to the experimental binding affinity. The total interaction energy (TIE) is described in Table 1. The minimum TIE was calculated as −67.209 kcal/mol, −52.369 kcal/mol, and −60.539 kcal/mol for Avn A, B, and C, respectively, and these binding poses were selected for PIEDA (Figure 3b–d). The PIEDA and PIE of the common residues among the selected binding poses are described in Appendix A and Figure 4. The common residues are defined as residues with a calculated absolute PIE of at least 4 kcal/mol in at least one Avn derivative. Of all the Avn derivatives, Asp73 is the strongest attractive residue among the other esterase residues and highly correlates with the result of the esterase inhibition assay. In addition, interaction with Lys284 was calculated to be attractive for only Avn B, but was repulsive for the other Avn derivatives. The difference in the interaction between Asp73/Lys284 and the other Avn derivatives is that the binding pose of Avn B was flipped compared to the other Avn derivatives (Figure 5). The nitrogen and carbonyl oxygen in the flipped structure of Avn B is closely located to the carbonyl oxygen in Asp73 and nitrogen in Lys284. The strong electrostatic interaction between oxygen and nitrogen can exist in Avn B, thereby giving rise to the possibility that it can exhibit high esterase inhibition activity.

### 2.5. Esterase Inhibition of Tamoxifen and Avenanthramide B

In order to confirm the predictions made by the various molecular models, an esterase inhibition assay was performed. As shown in Figure 6, among the different Avn derivatives, Avn B is the only one to show significant inhibitory action, similar to that of tamoxifen. This esterase inhibition assay result is in accordance with the prediction made by the various molecular models, thus confirming that Avn B has the potential to inhibit the action of PLE.

## 3. Discussion

This study shows that tamoxifen can inhibit PLE enzyme activity. Interestingly, it has been shown that PLE and tamoxifen could interact with each other through the similar binding pocket of HLE. Tamoxifen can now be defined as an esterase inhibitor. We have also shown that, among the many Avn derivatives, Avn B has inhibitory effects on PLE, with a similar level of inhibition to that of tamoxifen. Avn A and C do not seem to inhibit PLE due to their positions when they interact with PLE. This fact could have various implications for both pharmaceutical and industrial perspectives. First, in terms of pharmaceuticals, various classes of esterases play roles in drug processing. For instance, tamoxifen, which is used as a control in this study, is known to interact with human CES 1 [28]. This interaction can lower the efficacy of the drug and cause side effects [28,29]. Therefore, by providing Avn B to interact with esterase instead of tamoxifen, the efficacy of tamoxifen, a chemotherapeutic drug, might be increased. In addition, consuming oats, which are a good source of Avn derivatives, can be potentially beneficial for those who are taking drugs. Second, from the industrial perspective, antibiotic supplements are widely used in raising farm animals. This widespread use can give rise to various issues, from remnant drugs in our food sources to drug resistance in animals [7]. The liver esterases of pigs are responsible for hydrolyzing antibiotic supplements; therefore, if Avn B is given, which inhibits PLE, this hydrolysis of antibiotics might be interrupted. It has been suggested that Avns exert their bioavailable effects on rat hepatic and cardiac muscles following their oral consumption [30]. In the farming industry, oats and other Avn B-containing crops are used. However, the use of these crops might intervene with the removal of antibiotics from the bodies of farm animals, as Avn B can inhibit the action of esterase. Therefore, the effects of Avn B-containing crops need to be further studied in farm animals. In addition, porcine liver esterase 5 (PLE5) was selected for our simulations because it is the only experimentally resolved isoform of PLE, providing a well-defined structural framework for computational studies. Although the specific role of PLE5 in porcine metabolism is not fully understood, its availability in the structural data makes it an ideal candidate for exploring esterase-inhibitory interactions. Although we have demonstrated that Avn B inhibits PLE in an experimental result, the underlying molecular mechanism remains unclear. Our docking simulations and molecular dynamics results suggest a specific binding mode for Avn B; however, we lack direct structural evidence, such as co-crystal structures, to confirm these predictions. Further studies, such as X-ray crystallography or cryo-EM, would provide more definitive structural information about the PLE–Avn B complex.

## 4. Materials and Methods

### 4.1. Materials and Esterase Inhibition Assay

Esterase inhibition measures the inhibition rate of tamoxifen using *p*-nitrophenyl acetate and PLE [31]. The 4-nitrophenyl acetate was purchased from MP Biomedicals (Santa Ana, CA, USA). The PLE, tamoxifen, and Avn A, B, and C were purchased from Sigma-Aldrich (St. Louis, MI, USA). Tamoxifen (100 μM) and Avn A, B, and C (100 μM) were diluted to 50 μM. The reagents and enzymes were of analytical grade. The diluted tamoxifen and Avns were then mixed with 4-nitrophenyl acetate, a substrate, and the PLE enzyme. The OD level was measured by spectrophotometry with a 265 wavelength for 10 min.

### 4.2. Molecular Docking Simulation

For the PLE binding pocket, the position of tamoxifen bound in human liver esterase (HLE) (PDB ID: 1YA4) (https://pdbj.org/emnavi/quick.php?id=pdb-1ya4, accessed on 4 December 2024) [32] was used because no pocket information exists for PLE. The binding pocket was defined using superimposed binding sites from the tamoxifen bonded crystal structures of HLE. The structures of the target protein, PLE (PDB ID: 5FV4), were obtained from the Protein Data Bank (PDB) database (https://www.rcsb.org/, accessed on 4 December 2024). The 3D structures of the Avn derivatives, which were Avn A (PubChem CID: 5281157), Avn B (CID: 10087955), and Avn C (CID: 11723200), were downloaded from PubChem (https://pubchem.ncbi.nlm.nih.gov/, accessed on 4 December 2024). The docking simulations were performed with AutoDock Vina 1.1.2 [33] and displayed in the Chimera 1.14 program [34]. The grid size used in AutoDock Vina was 20.

### 4.3. FMO Calculation and PIEDA

In this work, the two-body FMO method was applied to all calculations for the FMO2/DFTB method. Since the molecular docking simulation result gives hydrogen-free structures, the Chimera program was used to add hydrogen atoms. All input files were prepared in compliance with the hybrid orbital projection scheme fragmentation. Two cysteine residues forming the disulfide bond were defined as one fragment. Other parameters calculated in the FMO calculation were default values. Total charges for tamoxifen and Avn derivatives were defined as zero. The FMO calculation and PIEDA were performed with the 30 June 2020 R1 version of GAMESS [35]. The pair interaction energy (PIE) was calculated to find significant esterase residues interacting with Avn derivatives. PIEDA can be used to analyze the composition of molecular interactions between two fragments using the decomposition method. The composition of the interactions in PIEDA are electrostatic interactions (ESs), exchange repulsion (EX), charge transfer (CT), dispersion (DI), and solvation (SL), which can consider electrostatic interaction, van der Waals interaction, hydrogen bond interaction, and solvation free energy.

### 4.4. Molecular Dynamics Simulation

To validate the binding poses of tamoxifen and three Avn derivatives, we performed molecular dynamics simulation using the GROMACS 2022.4 package [36]. We employed the CHARMM36m force field [37] combined with the CHARMM-modified TIP3P model. To improve charge–charge interaction pairs, we applied the CUFIX corrections to the CHARMM36m force field set [38]. The model structured was first minimized with steepest descent followed by maximum force under 100 kJ mol^−1^ nm^−1^. We performed the equilibrium calculation for the constant volume–temperature (NVT) and pressure–temperature (NPT) ensembles under 100 ps. For the computation of van der Waals forces, we employed a 10 to 12 Å switching scheme. We computed the long-range electrostatic forces using the particle-mesh Ewald summation scheme [39] with a 1.2 Å grid spacing and a 12 Å real-space cutoff. Covalent bonds to hydrogen in non-water and water molecules were constrained using the LINCS [40] and SETTLE [41] algorithms. The production was performed by the NPT ensemble under 300 ns. The interaction energy between PLE and each ligand was calculated by gmx_MMPBSA [42].

## 5. Conclusions

Avn B has been shown to have similar inhibitory effects to those of tamoxifen. Therefore, this study could serve as the fundamental basis for future studies to further investigate the complex role of Avns or tamoxifen in terms of liver esterase function, which could have both pharmaceutical and industrial agriculture implications. Also, the present study could contribute to understanding the potential threat of drugs intended for human use. In addition, this study could be used to develop a new animal model to study the hepatotoxicity of tamoxifen.

## Figures and Tables

**Figure 1 ijms-25-13291-f001:**
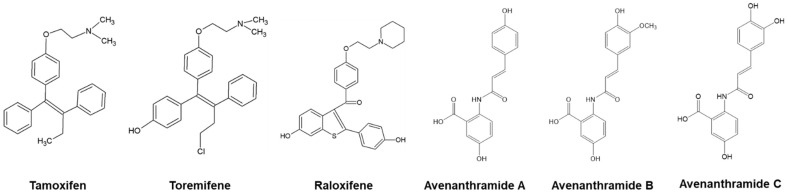
Structure of three different selective estrogen receptor modulators (tamoxifen, toremifene, and raloxifene) and avenanthramides. Tamoxifen and toremifene contain triphenylethylene, whereas raloxifene has a benzothiophene structure.

**Figure 2 ijms-25-13291-f002:**
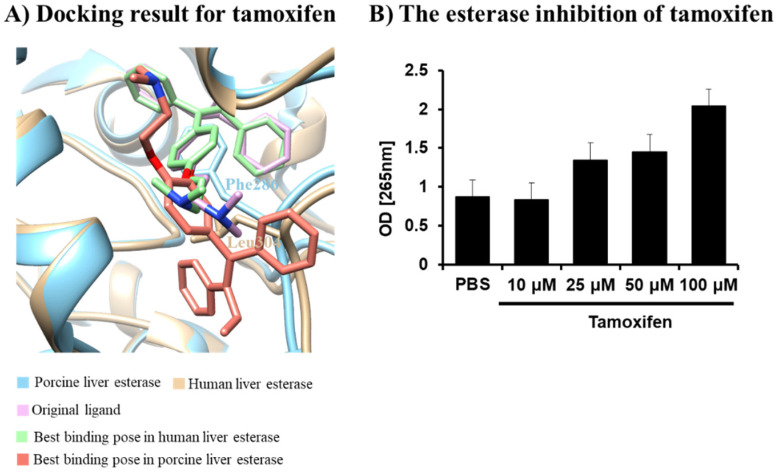
The superimposed structure between PLE and HLE (**A**) and the esterase inhibition by tamoxifen (**B**). The white structure represents tamoxifen. The tan and cyan structures represent HLE and PLE, respectively. The pink structure represents the crystal structure of tamoxifen. The salmon and green structures represent the best binding pose of tamoxifen in HLE and PLE. Tamoxifen is shown to inhibit the activity of porcine liver esterase dose-dependently.

**Figure 3 ijms-25-13291-f003:**
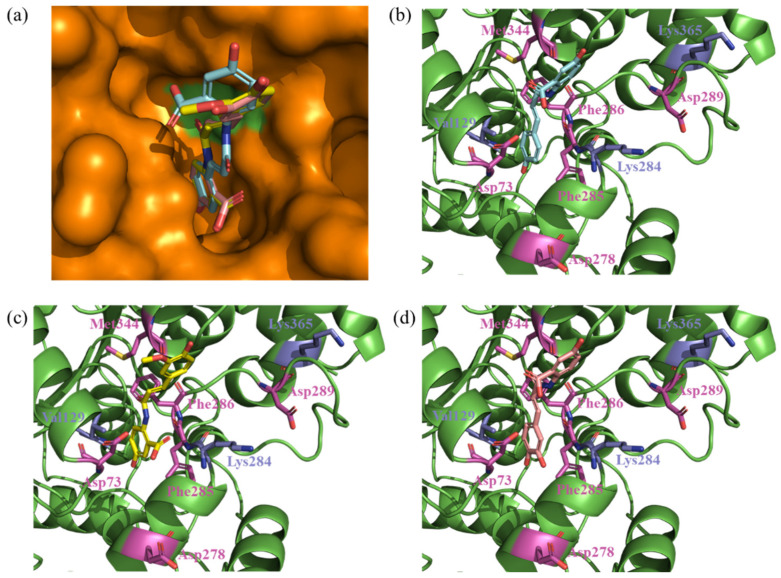
The selected binding poses of avenanthramide derivatives using molecular docking simulation. (**a**) All avenanthramide derivatives, with the orange surface representing the PLE. Each avenanthramide derivative was generated by molecular docking. (**b**) Avenanthramide A is represented by the blue structure, (**c**) avenanthramide B is represented by the yellow structure, and (**d**) avenanthramide C is represented by the pink structure. The orange surface represents the PLE. None of the binding poses of the avenanthramide derivatives could be located in the active site of PLE; instead, they were located outside of the active site because Phe286 was blocking the entrance to the active site of PLE. The green surface represents Phe286 in PLE.

**Figure 4 ijms-25-13291-f004:**
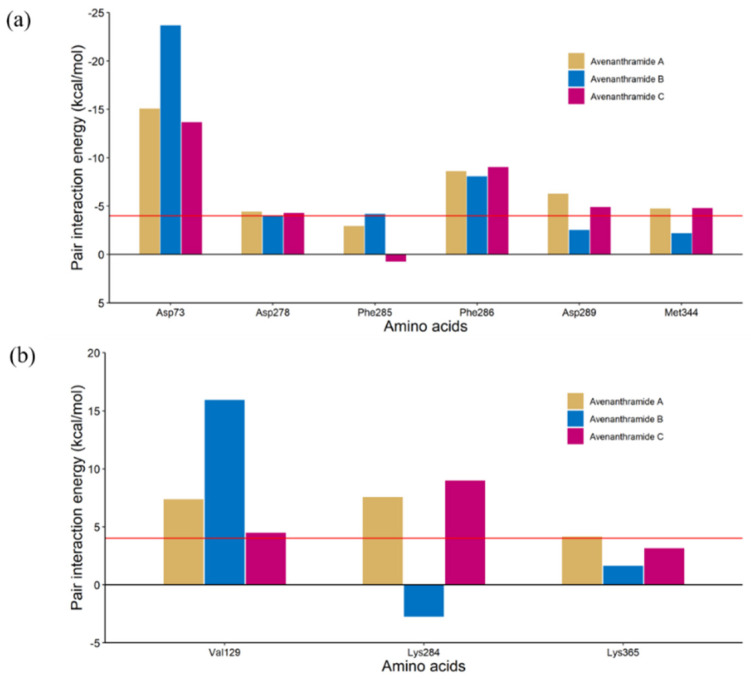
The PIE measures. (**a**) The PIE of the attractive common residue. (**b**) The PIE of the repulsive common residue. The red lines represent the minimum absolute PIE value (4 kcal/mol) for selection as a common residue.

**Figure 5 ijms-25-13291-f005:**
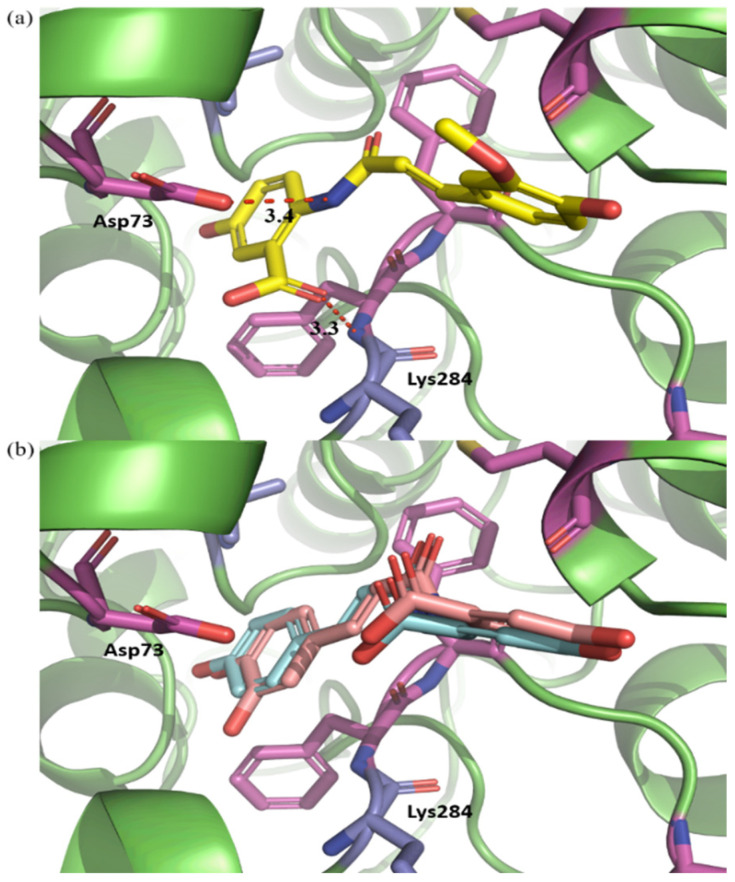
The interaction difference between (**a**) avenanthramide B and (**b**) avenanthramide A and C. The difference between (**a**) avenanthramide B and (**b**) avenanthramide A and C. The significant interactions between avenanthramide B and the other avenanthramide derivatives are represented by the red dashed line.

**Figure 6 ijms-25-13291-f006:**
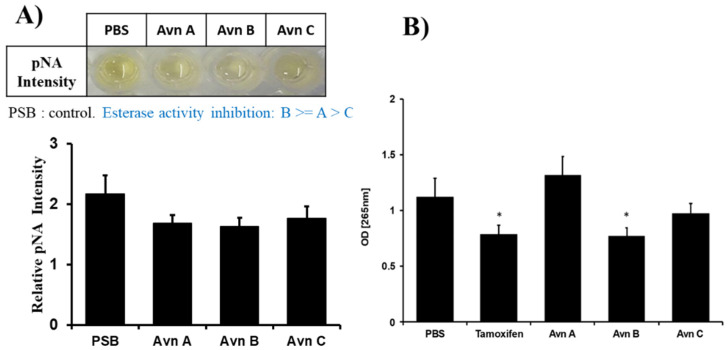
The comparison of the PLE-inhibitory activity of tamoxifen and avenanthramide A, B, and C. (**A**) Inhibition of porcine liver esterase activity by avenanthramide A, B, and C. (**B**) Comparison of porcine liver esterase-inhibitory activity of tamoxifen and avenanthramide A, B, and C. The statistically significant inhibition of PLE activity was induced by tamoxifen and avenanthramide B, but not avenanthramide A and C. The results shown are the mean ± SEM and represent three independent tests. * *p* < 0.05 = significant differences.

**Table 1 ijms-25-13291-t001:** The PIEDA of the common residues among the selected binding poses of each avenanthramide derivative.

Compound Name	Residue	PIE ^a^	Component of PIE (kcal/mol) ^b^				
			ES	EX	CT	DI	SL
Avenanthramide A	Attractive						
	Asp73	−15.070	−1.064	0.132	0.003	−4.273	−9.868
	Asp278	−4.420	−0.656	0.000	0.000	−0.056	−3.708
	Phe285	−2.940	0.128	0.420	−0.103	−3.618	0.233
	Phe286	−8.603	−3.960	0.195	−0.034	−4.984	0.180
	Asp289	−6.278	−3.981	0.000	0.000	−0.123	−2.174
	Met344	−4.753	−4.212	0.002	−0.003	−1.594	1.054
	Repulsive						
	Val129	7.359	−1.178	8.561	−0.081	−0.384	0.441
	Lys284	7.551	4.793	−0.001	0.000	−0.965	3.724
	Lys365	4.148	1.946	0.000	0.000	−0.010	2.212
Avenanthramide B	Attractive						
	Asp73	−23.682	−16.924	−0.413	−0.318	−3.728	−2.299
	Asp278	−4.026	−1.519	0.000	0.000	−0.076	−2.431
	Phe285	−4.199	0.303	0.575	−0.099	−4.407	−0.571
	Phe286	−8.080	−1.910	0.425	−0.119	−5.153	−1.323
	Asp289	−2.522	0.509	0.000	0.000	−0.137	−2.894
	Met344	−2.207	−0.570	0.000	0.000	−1.391	−0.246
	Repulsive						
	Val129	15.942	0.579	15.411	−0.133	−0.157	0.242
	Lys284	−2.752	−7.505	−0.021	−0.140	−1.138	6.052
	Lys365	1.630	−0.239	0.000	0.000	−0.010	1.879
Avenanthramide C	Attractive						
	Asp73	−13.672	−2.866	0.124	−0.034	−4.115	−6.781
	Asp278	−4.287	−1.987	0.000	0.000	−0.061	−2.239
	Phe285	0.741	1.122	1.934	0.245	−2.921	0.361
	Phe286	−9.027	−4.630	0.313	−0.043	−4.894	0.227
	Asp289	−4.893	−3.389	0.000	0.000	−0.104	−1.399
	Met344	−4.800	−3.821	0.001	−0.002	−1.744	0.766
	Repulsive						
	Val129	4.476	0.369	5.009	−0.079	−0.950	0.127
	Lys284	8.991	9.097	−0.006	0.000	−1.223	1.123
	Lys365	3.150	1.482	0.000	0.000	−0.009	1.677

^a^ PIE: Pair interaction energy between compound and esterase residue. ^b^ ES: Electrostatic interaction, EX: exchange repulsion, CT: charge transfer, DI: dispersion, SL: solvation.

## Data Availability

Data are contained within the article and Appendix A.

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
