# Peer review of "Compared Inhibitory Activities of Tamoxifen and Avenanthramide B on Liver Esterase and Correlation Based on the Superimposed Structure Between Porcine and Human Liver Esterase"

_ijms, 2024, doi:10.3390/ijms252413291_

Round 1

Reviewer 1 Report (Previous Reviewer 3)

Comments and Suggestions for Authors

 In this manuscript, the authors present computational and experimental results on the inhibitory activity of tamoxifen and several avenanthramides. The inhibitory activity of tamoxifen is well-known, and the authors use it as a control for the other inhibitor candidates. The aim of studying avenanthramides is to modulate the PLE activity and avoid tamoxifen side effects. Remarkably, the experimental results confirm the inhibitory activity of avenanthramide B.

The main problem is that this manuscript is not well written.

English writing should be revised throughout the manuscript. Some grammatical errors and sentences have no meaning.  Some examples:

All closest distance from Phe286 to each ligand were calculated under 3.2Å, which can be blocked the active site of PLE.  (page 3) This sentence must be rewritten.

Since the RMSDs of all ligand were calculated under the resolution of PLE, all binding poses of all ligand is stable pose, and Avn_B is highest binding affinity among Avn derivatives and Tamoxifen.  (page 3) In this sentence, there are grammatical errors that must be clarified.

The study performed molecular docking simulation to determine the structure of the complex of PLE protein and Avn derivatives. Change to: In this study, a molecular docking simulation was performed to determine … (page 3)

The complex number was predicted as six poses in Avn A and three poses in the other Avn derivatives, by the AutoDock program. What does “the complex number” mean? (page 3)

In addition, the mean scores of RMSD by molecular dynamics were Tamoxifen (0.128), Avn_B (0.092), Avn_C (0.103), and Avn_A (0.209)

The RMSDs’ values have units of distance. When calculating RMSDs, which atoms are included in the RMSD calculation must be specified. RMSD stands for Root Mean Square Deviation, not for Root Mean Squared Deviation. (page 3)

Tamoxifen could fit into the binding site of HLE and the superimposed PLE. (page 3)

This sentence must be rewritten to understand its meaning. If tamoxifen could fit in the PLE binding site, some binding solution of tamoxifen bound to the PLE active site should be explored. The sentence seems contradictory with the tamoxifen solutions located at the PLE pocket entry.

(b) blue structure is avenanthramide A described by blue structure, (Figure 3 caption). This sentence must be corrected.

The green surface represents Phe286 in PLE.  (Figure 3 caption) This green surface cannot be observed in Figure 3.

In this study, it is experimentally shown that tamoxifen inhibits PLE, but the molecular explanation for this inhibitory effect is speculative. The docking poses obtained for tamoxifen are found at the entrance of the active site of PLE. The MD trajectories show that those poses are stable, and the authors explain the inhibitory activity of tamoxifen by the hypothesis that tamoxifen might block the entrance of the substrate to the active site. However, this conclusion needs to be better described. Better plots must be included that show the entrance of the cavity in comparison with the binding site cavity. Moreover, the blockage of Phe286 should be tested by MD replicas initiated with different rotamers of this residue.

According to the calculations, all the binding poses of the Avn-derivatives could not be located in the active site of PLE but were found outside because Phe286 was blocking entrance to the active site of PLE. Even though MD simulations verify the stability of those poses, other orientations of Phe286 have not been analyzed.

Comments on the Quality of English Language

English writing must be revised.

Author Response

Reviewer-1

In this manuscript, the authors present computational and experimental results on the inhibitory activity of tamoxifen and several avenanthramides. The inhibitory activity of tamoxifen is well-known, and the authors use it as a control for the other inhibitor candidates. The aim of studying avenanthramides is to modulate the PLE activity and avoid tamoxifen side effects. Remarkably, the experimental results confirm the inhibitory activity of avenanthramide B.

The main problem is that this manuscript is not well written.

English writing should be revised throughout the manuscript. Some grammatical errors and sentences have no Comments and Suggestions for Authors

Authors have improved the manuscript according to the suggestion of the reviewers.

meaning.  Some examples:

All closest distance from Phe286 to each ligand were calculated under 3.2Å, which can be blocked the active site of PLE. (page 3) This sentence must be rewritten.

As your comments, we have revised the sentence clearly.

Revised sentence: “The shortest distances between Phe286 and each ligand were calculated to be below 3.2 Å. This distance could potentially occlude the active site of PLE.”

Since the RMSDs of all ligand were calculated under the resolution of PLE, all binding poses of all ligand is stable pose, and Avn_B is highest binding affinity among Avn derivatives and Tamoxifen.  (page 3) In this sentence, there are grammatical errors that must be clarified.

We apologize for the error. The sentence has been revised in order to ensure clarity.

Revised sentences: “The Avn_B was highest binding affinity among Avn derivatives and Tamoxifen. Given that RMSDs of all ligands were calculated under 2.4Å resolution corresponding to the experimental-defined structure of PLE (PDB ID: 5FV4), all binding poses of the ligands could be regarded as stable pose.”

The study performed molecular docking simulation to determine the structure of the complex of PLE protein and Avn derivatives. Change to: In this study, a molecular docking simulation was performed to determine … (page 3)

Thank you for your correction.

The complex number was predicted as six poses in Avn A and three poses in the other Avn derivatives, by the AutoDock program. What does “the complex number” mean? (page 3)

Sorry about confusing sentences. The word “complex number” was mis-written. We fix to “number of complex structure”.

In addition, the mean scores of RMSD by molecular dynamics were Tamoxifen (0.128), Avn_B (0.092), Avn_C (0.103), and Avn_A (0.209). The RMSDs’ values have units of distance. When calculating RMSDs, which atoms are included in the RMSD calculation must be specified. RMSD stands for Root Mean Square Deviation, not for Root Mean Squared Deviation. (page 3)

Sorry about the confusing sentences. In our analysis, the full word of RMSD is root mean squared distance not deviation, which described to “The docking scores for the Avn derivatives and root mean squared distance (RMSD) from the binding pose are listed in Supplemental Table 1.” in 2.2 section. Additionally, the RMSD was calculated based on initial ligand structure. Therefore, we fix to this sentence. In addition, we added Å as the unit of distance.

Revised sentences: “The docking scores for the Avn derivatives and root mean square distance (RMSD) from the binding pose are listed in Supplemental Table 1.” and “In addition, the average RMSD for all ligands generated by molecular dynamics relative to initial ligand structure were Tamoxifen (0.128 Å), Avn_B (0.092 Å), Avn_C (0.103 Å), and Avn_A (0.209 Å).”

Tamoxifen could fit into the binding site of HLE and the superimposed PLE. (page 3)

This sentence must be rewritten to understand its meaning. If tamoxifen could fit in the PLE binding site, some binding solution of tamoxifen bound to the PLE active site should be explored. The sentence seems contradictory with the tamoxifen solutions located at the PLE pocket entry.

The sentence is removed as it confuses the reader as to the result of the docking simulation.

(b) blue structure is avenanthramide A described by blue structure, (Figure 3 caption). This sentence must be corrected.

Thank you for your correction. We removed “blue structure is”.

Revised sentence: “(b) avenanthramide A described by blue structure.”

The green surface represents Phe286 in PLE.  (Figure 3 caption) This green surface cannot be observed in Figure 3.

Please be advised that Figure 3a contained erroneous information. The caption for Figure 3 has been amended to include a green surface.

Revised figure:

In this study, it is experimentally shown that tamoxifen inhibits PLE, but the molecular explanation for this inhibitory effect is speculative. The docking poses obtained for tamoxifen are found at the entrance of the active site of PLE.

The MD trajectories show that those poses are stable, and the authors explain the inhibitory activity of tamoxifen by the hypothesis that tamoxifen might block the entrance of the substrate to the active site. However, this conclusion needs to be better described. Better plots must be included that show the entrance of the cavity in comparison with the binding site cavity. Moreover, the blockage of Phe286 should be tested by MD replicas initiated with different rotamers of this residue.

We agree with your points that the molecular explanation for this inhibitory effect is speculative. We tried molecular dynamics to elucidate the obstruction of Avn through the utilisation of different rotamers of Phe286. However, the MD simulation could not be conducted due to the occurrence of crashes with neighboring atoms, which was attributed to the insufficient flexibility to form the Phe286 rotamer. In addition, we described our limitation in Discussion section.

Revised sentences: “Although we have demonstrated that AvnB inhibits PLE in an experimental result, the underlying molecular mechanism remains unclear. Our docking simulations and molecular dynamics results suggest a specific binding mode for Avn B, however, we lack direct structural evidence, such as co-crystal structures, to confirm these predictions. Further studies, such as X-ray crystallography or cryo-EM, would provide more definitive structural information about the PLE-Avn B complex.”

Reviewer 2 Report (Previous Reviewer 2)

Comments and Suggestions for Authors

Authors have improved the manuscript according to the suggestion of the reviewers.

Author Response

Reviewer-2

According to the calculations, all the binding poses of the Avn-derivatives could not be located in the active site of PLE but were found outside because Phe286 was blocking entrance to the active site of PLE. Even though MD simulations verify the stability of those poses, other orientations of Phe286 have not been analyzed.

It was difficulty in generating alternative orientations of Phe286 within the pocket of PLE, because of the occurrence of crashes between neighboring atoms and Phe286. Therefore, we described our limitation about computational structure analysis in Discussion section.

Revised sentences: “Although we have demonstrated that AvnB inhibits PLE in an experimental result, the underlying molecular mechanism remains unclear. Our docking simulations and molecular dynamics results suggest a specific binding mode for Avn B, however, we lack direct structural evidence, such as co-crystal structures, to confirm these predictions. Further studies, such as X-ray crystallography or cryo-EM, would provide more definitive structural information about the PLE-Avn B complex.”

Round 2

Reviewer 1 Report (Previous Reviewer 3)

Comments and Suggestions for Authors

In this version the authors have improved the manuscript answering to my comments.  The paper is publishable in the present form.

Comments on the Quality of English Language

English language still needs revision

Author Response

Although the submitted version of this manuscript takes into account what highlighted by the two reviewers according to the text pitfalls, some instances have still to be implemented

1) The dynamics simulation length, as well as the relative discussion in the body of the manuscript, is still too short and concise. I do suggest extending up to 1000 or at least 500 ns the MD runs

 :As your comments, we performed to append the length of the molecular dynamics simulation between PLE and tamoxifen, and Avn derivatives. Due to the deadline for revision, the 300 ns simulation was not completed to 1000 nm. The additional information has been included in Supplemental Figure 1. The results of the dynamics simulations during 300 ns supported the conclusion that all binding poses of the tamoxifen and Avn derivatives could be regarded as stable pose, as evidenced by a standard deviation of RMSD that is similar to that of the X-ray crystal structure of PLE (5FV4).

Revised sentences in the Result section: “In addition, the standard deviation of RMSD for all ligands generated by molecular dynamics relative to initial ligand structure were Tamoxifen (0.956 Å), Avn_A (2.357 Å), Avn_B (2.863 Å), and Avn_C (1.305 Å). Given that the standard deviation of RMSDs of all ligands were calculated similar to 2.4 Å resolution corresponding to the experimental-defined structure of PLE (PDB ID: 5FV4), all binding poses of the ligands could be regarded as stable pose.”

2) English language need to be checked by native language subject

: Thank you for your comment. We again check English language and references of manuscript.

This manuscript is a resubmission of an earlier submission. The following is a list of the peer review reports and author responses from that submission.

Round 1

Reviewer 1 Report

Comments and Suggestions for Authors

The manuscript presents an interesting topic from a scientific perspective. From my side, the article can be accepted with the following corrections:

- The authors referenced older works in the introduction, with the newest being from 2022. Since this is an interesting topic for the scientific community, the authors should include more recent references and discussions of results in the introduction.

- In section 2.1 of the materials and methods, the authors did not include the purity of the purchased materials. In interaction studies, high purity is expected. The authors should include this information.

- Why was the structure with a resolution of (5FV4) 2.4 Å chosen? Since it deals with molecular interactions, it would be interesting for the authors to attempt to reproduce the study with a higher resolution structure to verify if the interaction distances are the same.

- In section 2.2, the authors did not clarify if a blind docking was performed on the protein, nor were the criteria for selecting conformations clarified. The authors should include this information in the methodology. From my reading, it seems the authors chose based solely on energy scoring; did they consider the statistical criterion (the number of conformations per energy function)? The authors should add this information (example of an article with methodology considering statistical weight: doi.org/10.3390/molecules25122841).

- Since the manuscript deals with interactions, the authors should describe the important interactions for the molecular mechanism (examples doi.org/10.3390/biomedicines8120629, doi.org/10.3390/molecules25122841).

- The results are interesting but were merely descriptive. The authors should include interaction articles in the discussions and compare their results.

- The authors must add a conclusion section.

Comments on the Quality of English Language

 Minor editing of English language required

Reviewer 2 Report

Comments and Suggestions for Authors

Authors present their work on inhibitory activities of tamoxifen and avenanthramide B to liver esterase. Work is mainly computationally based with a small part of experimental determination of inhibitory activities. The rational of work is not clearly described which results in a questionable methodology used for the answering scientific question, most importantly “conclusions” are highly limited and of questionable interest to the readers of International Journal of Molecular Sciences.

I do not suggest publication.

Comments on the Quality of English Language

no special comment on grammar

Reviewer 3 Report

Comments and Suggestions for Authors

              In this manuscript, the authors present computational and experimental results on the inhibitory activity of tamoxifen and several avenanthramides. The inhibitory activity of tamoxifen is well-known. Therefore, the authors used it as a template. The aim of studying avenanthramides is to modulate the PLE activity and avoid tamoxifen side effects.

              The study presents valuable aspects. However, it also presents several relevant flaws that make it unpublishable in its present form.

1.      The conclusions concerning the structures of the different ligand/enzyme complexes are based on docking calculations. Docking is an low-level option because dynamical effects are not included in the simulations.

2.      The docking simulations are complemented by calculating ligand/receptor interactions at the QM level using FMO and PIEDA methods. The application of this methodology gives extra value to this study. However, the main problem here is that these QM calculations are done using the docking poses, and for the Avn-derivatives, all the binding modes are outside the PLE active site.

3.      In this study, it is experimentally shown that tamoxifen inhibits PLE, but the molecular explanation for this inhibitory effect is speculative. The docking poses obtained for tamoxifen are found at the entrance of the active site of PLE. The authors explain the inhibitory activity of tamoxifen by the hypothesis that tamoxifen might block the entrance of the substrate to the active site. This conclusion needs to be proven.

4.      The other problem in this study is that all the binding poses of the Avn-derivatives could not be located in the active site of PLE but were outside because Phe286 was blocking the entrance to the active site of PLE. An MD simulation could verify if other orientations of Phe286 are possible.

5.      Change “classical QM calculations” on page  3 to “ab initio QM calculations”

6.      The two RMSDs in Figure 2 must be explained.

Round 2

Reviewer 2 Report

Comments and Suggestions for Authors

Authors have improved the manuscript according to the suggestions. Newly added experimental data (Figure 7) are poorly presented. It is not clear what is the difference between Inhibition of porcine liver esterase activity by avenanthramide A, B, and C and comparison of porcine liver esterase-inhibitory activity by tamoxifen and avenanthramide A, B and C.

Figure B - measured quantity stated is OD which is unusual in the cases where you have pNA, in such cases you measure absorbance at 405 nm.

It should be also stated at what concentration it was tested ( (not only at experimental details) as 50 uM is quite high conc and relevance at such high concentrations should be taken into account.

Author Response

September 10, 2024

JIMS, Editorial Office

REF:  ijms-3133840

Title: Compared inhibitory activities of tamoxifen and avenanthramide B to liver esterase and correlation based on the superimposed structure between porcine and human liver esterase

Authors: Hakseong Lim, Sungbo Hwang, Seung-Hak Cho, Young-Seok Bak, Woong-suk, Yang, Daeui Park *, Cheorl-Ho Kim *

I would like to appreciate you and Reviewer-2 to 3rd-review our re-revised manuscript entitled "Compared inhibitory activities of tamoxifen and avenanthramide B to liver esterase and correlation based on the superimposed structure between porcine and human liver esterase" for publication in IJMS.

The 3rd review comments were still valuable and have been revisited to meet the IJMS criteria and standard for publication. Item by item revision has been made in the separate review answers.

(I) Ensure all references are relevant to the content of the manuscript.

Answer: Yes, the references have been rechecked to be relevant.

(II) Highlight any revisions to the manuscript, so editors and reviewers can see any changes made.

Answer: All highlights have been colored during revision.

(III) Provide a cover letter to respond to the reviewers’ comments and explain, point by point, the details of the manuscript revisions.

Answer: Yes, we have done, See below

(IV) If the reviewer(s) recommended references, critically analyze them to ensure that their inclusion would enhance your manuscript. If you believe these references are unnecessary, you should not include them.

Answer: Yes, no changes in references. Only one reference to respond the review has been described in the revision letter. It doesn’t need to insert due to similar references.

(V) If you found it impossible to address certain comments in the review reports, include an explanation in your appeal.

Answer: The reviewer’s comments are agreeable and answered. For the changes in the text during the current revision, reference arrangements are performed, as described:

Question: Authors have improved the manuscript according to the suggestions. Newly added experimental data (Figure 7) are poorly presented. It is not clear what is the difference between Inhibition of porcine liver esterase activity by avenanthramide A, B, and C and comparison of porcine liver esterase-inhibitory activity by tamoxifen and avenanthramide A, B and C.

Answer: The pNA-detection is not well differentiated by the optical visuals but the optical observation show differences between Ave A, AveB, AveC and PBS. Therefore, generally 265 nm detection is performed as we have done by intensity scanning tools in Fig. 2 and Fig. 7. This is a general method, as I cited and described below: Li G, Xu L, Zhang H, Liu J, Yan J, Yan Y. A De Novo Designed Esterase with p-Nitrophenyl Acetate Hydrolysis Activity. Molecules. 2020 Oct 13;25(20):4658. doi: 10.3390/molecules25204658.

Thus, the difference between inhibition of porcine liver esterase activity by avenanthramide A, B, and C and comparison of porcine liver esterase-inhibitory activity by tamoxifen and avenanthramide A, B and C are described. In conclusion, the Fig 7B shows the esterase-inhibitory activity intensity of the tamoxifen and avenanthramide A, B, and C.

Fig. 7.

Fig. 2.

Question: Figure B - measured quantity stated is OD which is unusual in the cases where you have pNA, in such cases you measure absorbance at 405 nm.

As suggested, the Figure B- measured quantity has been expressed as OD at 265 nm, but not at 405nm. The 365 nm is a specific adsorption wavelength.

Question: It should be also stated at what concentration it was tested (not only at experimental details) as 50 uM is quite high conc and relevance at such high concentrations should be taken into account.

As suggested, the present assays of the esterase enzyme activity have been performed using the pNA conc. 50 uM. For example, other article (I mentioned above: Li G, Xu L, Zhang H, Liu J, Yan J, Yan Y. A De Novo Designed Esterase with p-Nitrophenyl Acetate Hydrolysis Activity. Molecules. 2020 Oct 13;25(20):4658. doi: 10.3390/molecules25204658) has used 1 mL of the reaction system containing 10 μL of p-NPA (100 mM), final concentration of 100 uM. This is calculated to be 2-times higher than our present 50 uM. In summary, the present concentration 50 uM of substrate pNA is general. However, I would like to thank the reviewer for his(her) careful reading.

To enhance the revision, the sentence of “Tamoxifen (100 μM) and Avn A, B, and C (100 μM) were diluted to 50 μM” has been revised in 4. Materials and methods

4.1. Materials and esterase inhibition assay

I would be happy if you could accept.

Sincerely
